# ToAlign: Task-oriented Alignment for Unsupervised Domain Adaptation

**Guoqiang Wei**[1][*] **Cuiling Lan**[2][†] **Wenjun Zeng**[2] **Zhizheng Zhang**[2] **Zhibo Chen**[1][†]

[1] University of Science and Technology of China    [2] Microsoft Research Asia
wgq7441@mail.ustc.edu.cn  {culan,wezeng,zhizzhang}@microsoft.com
chenzhibo@ustc.edu.cn

## Abstract

Unsupervised domain adaptive classification intends to improve the *classification* performance on unlabeled target domain. To alleviate the adverse effect of domain shift, many approaches align the source and target domains in the feature space. However, a feature is usually taken as a whole for alignment without explicitly making domain alignment proactively serve the classification task, leading to sub-optimal solution. In this paper, we propose an effective ***Task-oriented Alignment*** (*ToAlign*) for unsupervised domain adaptation (UDA). We study what features should be aligned across domains and propose to make the domain alignment proactively serve classification by performing feature decomposition and alignment under the guidance of the prior knowledge induced from the classification task itself. Particularly, we explicitly decompose a feature in the source domain into a task-related/discriminative feature that should be aligned, and a task-irrelevant feature that should be avoided/ignored, based on the classification meta-knowledge. Extensive experimental results on various benchmarks (*e.g.*, Office-Home, Visda-2017, and DomainNet) under different domain adaptation settings demonstrate the effectiveness of *ToAlign* which helps achieve the state-of-the-art performance. The code is publicly available at https://github.com/microsoft/UDA.

## 1   Introduction

Convolutional Neural Networks (CNNs) have made extraordinary progress in various computer vision tasks, with image classification as a most representative one. The trained models generally perform well on the testing data which shares similar data distribution to that of the training data. However, in many practical scenarios, drastic performance degradation is observed when applying such trained models to new domains with *domain shift* [57], where the data distributions between the training and testing domains are different. Fine-tuning on labeled target data is a direct solution but is costly due to the requirement of target sample annotations. In contrast, unsupervised domain adaptation (UDA) requires only the labeled source data and *unlabeled* target data to enhance the model's performance on the target domain, which has attracted increasing interest in both academia [3, 2, 72, 58, 25, 31] and industry [63, 27].

There has been a large spectrum of UDA methods. Supported by the theoretical analysis [3], the overwhelming majority of methods tend to align the distributions of source and target domains. A line of works [6, 67, 43, 54, 55] explicitly align the distributions based on domain discrepancy measurements, *e.g.*, Maximum Mean Discrepancy (MMD) [6]. Another line of alignment-based UDAs borrow ideas from Generative Adversarial Networks [19] and use domain adversarial training

---

[*]This work was done when Guoqiang Wei was an intern at MSRA.
[†]Corresponding author.

35th Conference on Neural Information Processing Systems (NeurIPS 2021).

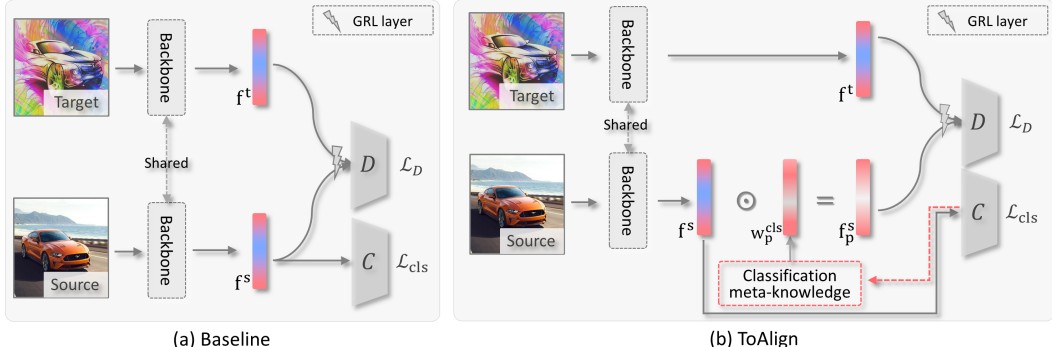

Figure 1: **Illustration of adversarial learning based (a) _Baseline_ and (b) our proposed _ToAlign_.**
$D$ and $C$ denote domain discriminator and image classifier respectively. (a) _Baseline_ (_e.g._, DANN
[18]) directly aligns the target feature $\mathbf{f}^t$ with the holistic source feature $\mathbf{f}^s$. Domain alignment and
image classification tasks are optimized in parallel. (b) Our proposed _ToAlign_ makes the domain
alignment proactively serve the classification task, where target feature $\mathbf{f}^t$ is aligned with source
task-discriminative "positive" feature $\mathbf{f}^s_p$ which is obtained under the guidance of meta-knowledge
induced from the classification task.    denotes Hadamard product.

to learn domain-aligned/invariant features, which dominate in the top performance methods. In the
seminal work Domain Adversarial Neural Network (DANN) [17, 18], a domain discriminator is
trained to distinguish the target features from source features while a feature extractor (generator) is
trained to generate domain-invariant features to fool this discriminator. Following DANN, a plethora
of variants have been proposed [58, 40, 51, 11, 50, 61, 38, 41, 14, 10, 64].

It is noteworthy that the goal of alignment in UDA is to alleviate the adverse effect of domain shift to
improve the _classification_ performance on unlabeled target data. Even though impressive progress has
been made, there is a common intrinsic limitation, _i.e._, **alignment is still not deliberately designed
to dedicatedly/proactively serve the final image classification task**. In many previous UDAs, as
shown in Figure 1 (a), the alignment task is in parallel with the ultimate classification task. The
assumption is that learning domain-invariant features (via alignment) reduces the domain gap and thus
makes the image classifier trained on source readily applicable to target [3]. However, with alignment
treated as a parallel task, there is a lack of mechanism to make it explicitly assist classification, where
the alignment may contaminate the discriminative features for classification [28]. Previous works
(_e.g._, CDAN [40]) exploit class information (_e.g._, predicted class probability) as a condition to the
discriminator. MADA [42] implements class-level domain alignment by applying one discriminator
per class. Their purpose is to provide additional helpful information to the discriminator [40] or
perform class-level alignment [42], but they are still short of explicitly making alignment assist
classification.

Some works move a step forward and investigate what features the networks should align for better
adaptation. [62, 32] focus on transferable local regions, which are selected based on the uncertainty
or entropy of the domain discriminator, for alignment. However, such self-induced feature selection
is still not specific to the optimization of classification task; instead, it is based on the alignment task
itself. There is no guarantee that alignment positively serves the classification task. Hsu _et al._ [23]
carry out object centerness-aware alignment by aligning the center part of the objects to exclude the
background distraction/noise for domain adaptive object detection. However, the feature in object
center position could be task-irrelevant and thus is not suited for alignment. Moreover, regarding such
centerness feature as alignment objective is somewhat ad-hoc, which is still not designed directly
from the perspective of assisting classification.

_We pinpoint that the selection of "right" features to achieve task-oriented alignment is important._ For
classification, the essence is to train the network to extract class-discriminative feature. Similarly, for
UDA classification, it is also desired to assure strong discrimination of the target domain features
without class label supervision. Thus, we intend to align target features to the task-discriminative
source features while ignoring the task-irrelevant ones. Note that for the feature of a source sample, it
contains both task/classification-discriminative and task-irrelevant information, because the network
is in general not able to suppress non-discriminative feature responses (_e.g._, responses unrelated to

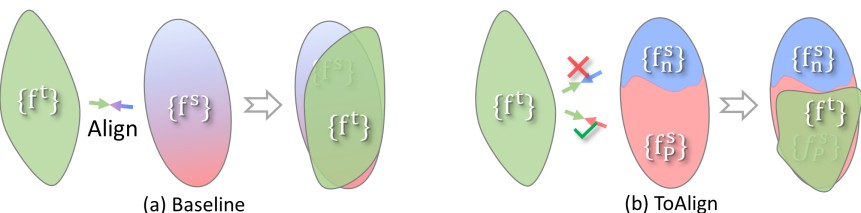

Figure 2: Conceptual comparison between (a) previous alignment and (b) our proposed task-oriented alignment. $\{\mathbf{f}^t\}$ and $\{\mathbf{f}^s\}$ denote the sets of target features and source features, respectively. (a) Previous methods take each source feature as a holistic one for alignment with target features. (b) We decompose each source feature $\mathbf{f}^s$ into a task-discriminative positive feature $\mathbf{f}_p^s$ and a task-irrelevant negative feature $\mathbf{f}_n^s$ and make the target features to be aligned with the positive source features $\{\mathbf{f}_p^s\}$ while avoiding aligning with the negative source features $\{\mathbf{f}_n^s\}$.

image class or those related to other tasks such as alignment) perfectly [52, 9]. Aligning target features with task-irrelevant source features would prevent alignment from serving classification and lead to poor adaptation. Intuitively, for example, image style that is a non-causal factor for classification can be considered as task-irrelevant information and the bias towards such factor in alignment may hurt the classification task. We demonstrate this by conducting experiments where only the source **t**ask-**i**rrelevant features are utilized to align with target *i.e.*, the scheme *Baseline+TiAlign* in Figure 3. The performance of *Baseline+TiAlign* (in purple) on target test set drops drastically compared to the source-only method which dose not incorporate any alignment technique. This corroborates that aligning with task-irrelevant features is even harmful to the classification on target domain.

Motivated by this, in this paper, we propose an effective UDA method named ***T**ask-oriented **A**lign*ment (*ToAlign*) to make the domain alignment explicitly serve classification. We achieve this by performing feature alignment guided by the meta-knowledge induced from the classification task to make the target features align with task-discriminative source features (*i.e.*, "positive" features), to avoid the interference from task-irrelevant features (*i.e.*, "negative" features). Figure 2 conceptually illustrates the comparison between our proposed alignment and previous one. Particularly, as illustrated in Figure 1 (b), to obtain the suitable feature from a source sample for alignment with target samples, we leverage the classification task to guide the extraction/distillation of task-related/discriminative feature $\mathbf{f}_p^s$, from original feature $\mathbf{f}^s$. Correspondingly, for the domain alignment task, we enforce aligning target features with the source positive features by domain adversarial training to achieve task-oriented alignment. In this way, the domain alignment will better assist the classification task.

We summarize our main contributions as follows:

- We pinpoint that the selection of "right" features to achieve task-orientated alignment is important for adaptation.
- We propose an effective UDA approach named ***ToAlign*** which enables the alignment to explicitly serve classification. We decompose a source feature into a task-relevant/discriminative one and a task-irrelevant one under the guidance of classification-meta knowledge for performing classification-oriented alignment, which explicitly guides the network what features should be aligned.

Extensive experimental results demonstrate the effectiveness of *ToAlign*. *ToAlign* is generic and can be applied to different adversarial learning based UDAs to enhance their adaption capability, which helps achieve the state-of-the-art performance with a negligible increase in training complexity and no increase in inference complexity.

## 2 Related Work

**Unsupervised Domain Adaptation** aims to transfer the knowledge from labeled source domain(s) to unlabeled target domain. Ben *et al.* [3] theoretically reveal that learning domain-invariant representations helps make the image classifier trained on source domain applicable to target domain. Various works learn domain-invariant features by aligning the source and target distributions measured by some metrics [6, 67, 43, 54, 55, 45], or by domain adversarial learning [58, 40, 51, 11, 50, 61, 38, 69, 56, 41, 14, 10, 64, 8, 29]. The latter is overwhelmingly popular in recent years owing to its

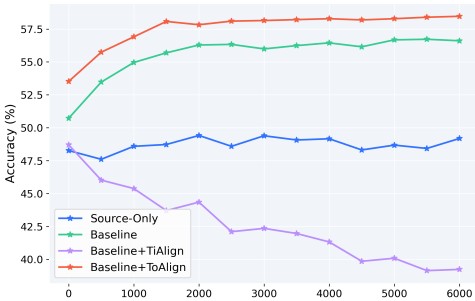

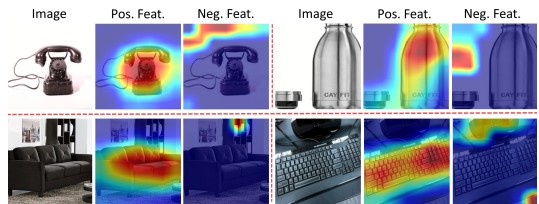

Figure 3: Classification accuracy on target (Rw→Cl in Office-Home) for different methods. *TiAlign* denotes aligning target features with **t**ask-**i**rrelevant source features.

Figure 4: Visualization of task-discriminative and task-irrelvant features. The positive features generally focus on the foreground objects which provide the most discriminative information for classification, while the negative ones focus on non-discriminative background regions. The images are sampled from Office-Home.

superiority in dealing with distribution problems [19]. Note that our proposed method is designed to enhance the capability of the widely used domain adversarial learning based approaches.

For domain adversarial learning based approach (*e.g.*, DANN [17, 18]), in general, a domain discriminator is trained to distinguish the source domain from the target domain, meanwhile a feature extractor is trained to learn domain-invariant features. Many variants of DANN have been proposed [40, 14, 56, 10, 12, 56, 69, 35, 5]. CDAN [40] further conditions the discriminator on the image class information conveyed in the classifier predictions. MADA [42] implements class-wise alignment with multi-discriminators. GSDA [24] performs class-, group- and domain-wise alignments simultaneously, where the three types of alignment are enforced to be consistent in their gradients for more precise alignment. HDA [12] leverages domain-specific representations as heuristics to obtain domain-invariant representations from a heuristic search perspective. CMSS [66] exploits Curriculum Learning (CL) [4] to align target samples with the dynamically selected source samples to exploit the different transferability of the source samples.

However, in these methods, the domain alignment is designed as a task in parallel with the image classification task. It does not explicitly take serving classification as its mission, where such alignment may result in loss of discriminative information. Jin *et al.* [28] remedy the loss of discriminative information caused by alignment via incorporating a restoration module. Wei *et al.* [64] pinpoint that alignment and classification are not well coordinated in optimization where they may contradict with each other. They thus propose to use meta-learning to coordinate their optimization directions.

In this paper, to make alignment explicitly serve classification, we propose a task-oriented alignment. Guided by the classification meta-knowledge, task-discriminative sub-features are selected for alignment. Different from [64], we investigate what features should be aligned to assist classification and intend to provide more interpretable alignment. We are the first to perform *task-oriented alignment* by decomposing source feature into task-discriminative and task-irrelevant feature, and explicitly guides the network what sub-features should be aligned. Note that Huang *et al.* [26] propose to decouple features into domain-invariant and domain-specific features, where the former ones are aligned for unsupervised person re-identification. [44, 7] exploit the VAE framework with several complex losses to perform the disentanglement from the perspective of domain and semantics simultaneously, and only use domain-invariant semantics for inference, leaving domain-specific but task-related information underexplored. In contrast to focusing on the domain-level, our decomposition strategy focuses on the task-level guided by image classification task, where we further enable domain alignment on the task-discriminative features to proactively serve image classification.

## 3 Task-Oriented Alignment for UDA

Unsupervised domain adaptation (UDA) for classification aims to train a classification model on labeled source domain image set $\mathbf{X}_s$ and unlabeled target domain image set $\mathbf{X}_t$ to obtain high classification accuracy on a target domain test set.

Most popular adversarial learning based UDAs attempt to align the features of the source and target domains to alleviate the domain gap to improve the classification performance on target domain. As

mentioned before, aligning based on holistic features is sub-optimal, where such alignment does not explicitly serve classification. To address this, as illustrated in Figure 1 (b), we propose an effective task-oriented alignment to explicitly make the alignment serve classification. Particularly, we propose to decompose a source sample feature into a task-discriminative one that should be aligned, and a task-irrelevant one that should be ignored based on the classification meta-knowledge. *Then, we perform alignment between the target features and the positive source features, which is consistent with the essence of the classification task, i.e., focusing on discriminative features.*

In Sec. 3.1, to be self-contained, we briefly introduce adversarial learning based UDAs. We answer the question of what feature should be aligned to better serve classification and introduce our task-oriented feature decomposition and alignment in Sec. 3.2.

## 3.1 Recap of Domain Adversarial UDAs

Domain adversarial learning based UDAs typically train a domain discriminator $D$ to distinguish which domain (*i.e.*, source or target) a sample belongs to, and adversarially train a feature extractor $G$ to fool the discriminator $D$ in order to learn domain-invariant feature representations. The network is also trained under the supervision of image classification on the labeled source samples. Particularly, $D$ is optimized to minimize the domain classification loss $\mathcal{L}_D$ (*i.e.*, binary cross entropy loss). Meanwhile, $G$ is optimized to maximize the domain classification loss $\mathcal{L}_D$ and minimize the image classification loss $\mathcal{L}_{cls}$ (*i.e.*, cross entropy loss):

$$\begin{aligned} &\underset{D}{\arg\min}\, \mathcal{L}_D, \\ &\underset{G}{\arg\min}\, \mathcal{L}_{cls} - \mathcal{L}_D, \end{aligned} \tag{1}$$

To achieve adversarial training, usually, gradient reversal layer (GRL) [17, 18] which connects $G$ and $D$ is used via multiplying the gradient from $D$ by a negative constant during the back-propagation to $G$. $\mathcal{L}_D$ is typically defined as [18, 40, 14]:

$$\mathcal{L}_D(\mathbf{X}_s, \mathbf{X}_t) = -\mathbb{E}_{\mathbf{x}_s \sim \mathbf{X}_s} \left[ \log(D(G(\mathbf{x}_s))) \right] - \mathbb{E}_{\mathbf{x}_t \sim \mathbf{X}_t} \left[ \log(1 - D(G(\mathbf{x}_t))) \right], \tag{2}$$

## 3.2 Task-oriented Feature Decomposition and Alignment

In adversarial learning based UDAs, a feature ingested by $D$ as a holistic feature from a source or target sample, in general contains both task/classification-discriminative information and task-irrelevant information. Intuitively, aligning the task-irrelevant features would not effectively reduce the domain gap of the task-discriminative features and thus brings no obvious benefit for the classification task. **Mistakenly aligning the target features with the source task-irrelevant features would hurt the discrimination power of the target features**. We also experimentally confirm that in Figure 3, *i.e.*, aligning with task-irrelevant features (*TiAlign*, line in purple) drastically reduces the classification accuracy on the target domain. **Therefore, we propose to decompose a holistic feature of each source sample into a task-discriminative feature and a task-irrelevant feature to enable the task-oriented alignment with the target features.**

Particularly, we softly select/re-weight (based on Grad-CAM [52]) the feature vector $\mathbf{f}^s$ of a source sample to obtain task-discriminative feature $\mathbf{f}_p^s$ that is discriminative for identifying the groundtruth class, which we refer to as *positive* feature. Correspondingly, the task-irrelevant feature $\mathbf{f}_n^s$ can be obtained simultaneously, which we refer to as *negative* feature.

**Task-Oriented Feature Decomposition.** Grad-CAM [73, 52, 9] is a widely used technique to localize the most important features for classification in a convolutional neural network model. As analyzed in [73, 52, 9, 53], the gradients (*w.r.t.* the feature for classification) of the final predicted score corresponding to the ground-truth class convey the task-discriminative information, which identifies the relevant features to recognize the image class correctly. It is noteworthy that such task-discriminative information is, in general, highly related (but not limited) to the foreground object in the classification task. *In this work, motivated by Grad-CAM, we propose to use the gradients of the predicted score corresponding to the ground-truth class as the attention weights to obtain the task-discriminative features.*

As illustrated in Figure 1, we obtain a feature map $F \in \mathbb{R}_+^{H \times W \times M}$ (*i.e.*, a tensor of non-negative real numbers, with height $H$, width $W$, and $M$ channels) from the final convolutional block (with ReLU

layer) of the feature extractor. After spatial-wise global average pooling (GAP), we have a feature vector $\mathbf{f} = pool(F) \in \mathbb{R}^M$. The logits for all classes are predicted via the classifier $C(\cdot)$. Based on the response $C(\mathbf{f})$, we can derive the gradient $\mathbf{w}_{cls} \in \mathbb{R}^M$ of $y^k$ w.r.t. $\mathbf{f}$:

$$\mathbf{w}^{cls} = \frac{\partial y^k}{\partial \mathbf{f}}, \tag{3}$$

where $y^k$ is the predicted score corresponding to the ground-truth class $k$. As analyzed in [52, 9, 53], the gradient $\mathbf{w}^{cls}$ conveys the channel-wise importance information of feature $\mathbf{f}$ for classifying the sample into its groudtruth class $k$. We draw inspiration from Grad-CAM which uses $\mathbf{w}^{cls}$ to modulate the feature map in channel-wise to find the classification-discriminative features. Similarly, modulated with $\mathbf{w}^{cls}$, we can obtain the task-discriminative (*i.e.*, positive) feature as:

$$\mathbf{f}_p = \mathbf{w}_p^{cls} \quad \mathbf{f} = s\mathbf{w}^{cls} \quad \mathbf{f}, \tag{4}$$

where represents the Hadamard product, the attention weight vector $\mathbf{w}_p^{cls} = s\mathbf{w}^{cls}$, where $s \in \mathbb{R}_+$ is an adaptive non-negative parameter to modulate the energy $\mathcal{E}(\mathbf{f}_p) = ||\mathbf{f}_p||_2^2$ of $\mathbf{f}_p$ such that $\mathcal{E}(\mathbf{f}_p) = \mathcal{E}(\mathbf{f})$:

$$s = \sqrt{\frac{||\mathbf{f}||_2^2}{||\mathbf{w}^{cls} \quad \mathbf{f}||_2^2}} = \sqrt{\frac{\sum_{m=1}^M f_m^2}{\sum_{m=1}^M (w_m^{cls} f_m)^2}}, \tag{5}$$

Motivated by the counterfactual analysis in [52], the task-irrelevant (*i.e.*, negative) feature can be represented as $\mathbf{f}_n = -\mathbf{w}_p^{cls} \quad \mathbf{f}$, where $-\mathbf{w}_p^{cls}$ delights the task-discriminative channels since the task-discriminative channels (with larger values in $\mathbf{w}_p^{cls}$) correspond to ones with smaller values in -$\mathbf{w}_p^{cls}$.

To better understand and validate the discriminativeness of the positive and negative features, we visualize the spatial maps $F$ with channels modulated by $\mathbf{w}^{cls}$ and $-\mathbf{w}^{cls}$ following [52, 73]. As shown in Figure 4, the positive information is more related to the foreground objects that provide the discriminative information for the classification task, while the negative one is more in connection with the non-discriminative background regions.

**Task-oriented Domain Alignment.** As discussed above, we expect the domain alignment to explicitly serve the final classification task. Given the source task-discriminative features obtained based on the classification meta-knowledge, we can guide the target features to be aligned with the source task-discriminative features $\mathbf{f}_p$ through different domain adversarial learning based alignment methods [17, 18, 12]. The procedure is almost the same as that in UDAs discussed in Sec. 3.1, except that the input source feature $\mathbf{f}^s$ to the final domain discriminator is replaced by the positive feature $\mathbf{f}_p^s$ of this source sample. Thus, the domain classification loss is defined with a small modification on Eq. (2):

$$\mathcal{L}_D(\mathbf{X}_s, \mathbf{X}_t) = -\mathbb{E}_{\mathbf{x}_s \sim \mathbf{X}_s} [\log(D(G^p(\mathbf{x}_s)))] - \mathbb{E}_{\mathbf{x}_t \sim \mathbf{X}_t} [\log(1 - D(G(\mathbf{x}_t)))]. \tag{6}$$

where $G^p(\mathbf{x}_s) = \mathbf{f}_p^s$ denotes the positive feature of source $\mathbf{x}_s$.

**Understanding from the Meta-knowledge Perspective.** To enable a better understanding of *ToAlign* on why it works well, here, we analyse *ToAlign* from the perspective of meta-learning with meta-knowledge.

In an adversarial UDA framework, the image classification task and domain alignment task can be considered to be a *meta-train* task $\mathcal{T}^{tr}$ and a *meta-test* task $\mathcal{T}^{te}$, respectively. *ToAlign* actually introduces knowledge communication from $\mathcal{T}^{tr}$ to $\mathcal{T}^{te}$. In the meta-training stage, we can obtain the prior/meta-knowledge $\phi^{tr}$ of $\mathcal{T}^{tr}$. Without effective communication between $\mathcal{T}^{tr}$ and $\mathcal{T}^{te}$, the optimization of $\mathcal{T}^{te}$ may contradict that of $\mathcal{T}^{tr}$, considering that they have different optimization goals. To improve the knowledge communication from $\mathcal{T}^{tr}$ to $\mathcal{T}^{te}$, certain meaningful prior/meta-knowledge $\phi^{tr}$ is helpful for a more effective $\mathcal{T}^{te}|_{\phi^{tr}}$. A typical implementation of passing meta-knowledge from $\mathcal{T}^{tr}$ to $\mathcal{T}^{te}$ is based on gradients [39, 16, 34, 64, 33], *i.e.*, $\nabla \mathcal{T}^{tr}$, which provides knowledge of $\mathcal{T}^{tr}$. Other mechanisms *e.g.*, leveraging the parameters regularizer in a way of weight decay, are also exploited [1, 71]. In our *ToAlign*, instead of encoding the meta-knowledge $\phi^{tr}$ into the gradients *w.r.t.* the parameters, we use $\mathcal{T}^{tr}$ to learn/derive attention weights for identifying $\mathcal{T}^{tr}$-related sub-features in the feature space and then pass such prior/meta-knowledge $\phi^{tr}$ to $\mathcal{T}^{te}$ to make meta-test task $\mathcal{T}_{\phi^{tr}}^{te}$ adapt its optimization based on $\phi^{tr}$.

| Method | Ar→Cl | Ar→Pr | Ar→Rw | Cl→Ar | Cl→Pr | Cl→Rw | Pr→Ar | Pr→Cl | Pr→Rw | Rw→Ar | Rw→Cl | Rw→Pr | Avg |
|---|---|---|---|---|---|---|---|---|---|---|---|---|---|
| Source-Only [21] | 34.9 | 50.0 | 58.0 | 37.4 | 41.9 | 46.2 | 38.5 | 31.2 | 60.4 | 53.9 | 41.2 | 59.9 | 46.1 |
| MCD(CVPR'18) [50] | 48.9 | 68.3 | 74.6 | 61.3 | 67.6 | 68.8 | 57.0 | 47.1 | 75.1 | 69.1 | 52.2 | 79.6 | 64.1 |
| CDAN(NeurIPS'18) [40] | 50.7 | 70.6 | 76.0 | 57.6 | 70.0 | 70.0 | 57.4 | 50.9 | 77.3 | 70.9 | 56.7 | 81.6 | 65.8 |
| ALDA(AAAI'20) [10] | 53.7 | 70.1 | 76.4 | 60.2 | 72.6 | 71.5 | 56.8 | 51.9 | 77.1 | 70.2 | 56.3 | 82.1 | 66.6 |
| SymNet(NeurIPS'18) [69] | 47.7 | 72.9 | 78.5 | 64.2 | 71.3 | 74.2 | 63.6 | 47.6 | 79.4 | 73.8 | 50.8 | 82.6 | 67.2 |
| TADA(AAAI'19) [62] | 53.1 | 72.3 | 77.2 | 59.1 | 71.2 | 72.1 | 59.7 | 53.1 | 78.4 | 72.4 | 60.0 | 82.9 | 67.6 |
| MDD(ICML'19) [68] | 54.9 | 73.7 | 77.8 | 60.0 | 71.4 | 71.8 | 61.2 | 53.6 | 78.1 | 72.5 | 60.2 | 82.3 | 68.1 |
| BNM(CVPR'20) [13] | 56.2 | 73.7 | 79.0 | 63.1 | 73.6 | 74.0 | 62.4 | 54.8 | 80.7 | 72.4 | 58.9 | 83.5 | 69.4 |
| GSDA(CVPR'20) [24] | **61.3** | 76.1 | 79.4 | 65.4 | 73.3 | 74.3 | 65.0 | 53.2 | 80.0 | 72.2 | 60.6 | 83.1 | 70.3 |
| GVB(CVPR'20) [14] | 57.0 | 74.7 | 79.8 | 64.6 | 74.1 | 74.6 | 65.2 | 55.1 | 81.0 | 74.6 | 59.7 | 84.3 | 70.4 |
| E-Mix(AAAI'21) [72] | 57.7 | 76.6 | 79.8 | 63.6 | 74.1 | 75.0 | 63.4 | 56.4 | 79.7 | 72.8 | **62.4** | **85.5** | 70.6 |
| MetaAlign(CVPR'21) [64] | 59.3 | 76.0 | 80.2 | 65.7 | 74.7 | 75.1 | 65.7 | 56.5 | 81.6 | 74.1 | 61.1 | 85.2 | 71.3 |
| DANNP [64] | 54.2 | 70.0 | 77.6 | 62.3 | 72.4 | 73.1 | 61.3 | 52.7 | 80.0 | 72.0 | 56.8 | 83.1 | 67.9 |
| DANNP+ToAlign | 56.8↑ | 74.8↑ | 79.9↑ | 64.0↑ | 73.9↑ | 75.3↑ | 63.8↑ | 53.7↑ | 81.1↑ | 73.1↑ | 58.2↑ | 84.0↑ | 69.9↑ |
| HDA(NeurIPS'20) [12] | 56.8 | 75.2 | 79.8 | 65.1 | 73.9 | 75.2 | 66.3 | 56.7 | 81.8 | **75.4** | 59.7 | 84.7 | 70.9 |
| HDA+ToAlign | 57.9↑ | **76.9**↑ | **80.8**↑ | **66.7**↑ | **75.6**↑ | **77.0**↑ | **67.8**↑ | **57.0**↑ | **82.5**↑ | 75.1↓ | 60.0↑ | 84.9↑ | **72.0**↑ |

Table 1: Accuracy (%) of different UDAs on Office-Home with ResNet-50 as backbone. Best in bold.

In this work, actually, we are motivated by the reliable human prior knowledge on *what* should be aligned across domains to better assist classification task for UDA (*i.e.*, task/classification-discriminative features), while excluding the interference from task-irrelevant ones. Accordingly, in our design, we obtain the prior/meta-knowledge for identifying task-discriminative features from the classification task (meta-train) and apply it to the domain alignment task (meta-test) to achieve task-*oriented* alignment.

# 4 Experiments

To evaluate the effectiveness of *ToAlign*, we conduct comprehensive experiments under three domain adaptation settings, *i.e.*, single source unsupervised domain adaptation (SUDA), multi-source unsupervised domain adaptation (MUDA) and semi-supervised domain adaptation (SSDA). For SSDA, domain adaptation is performed from labeled source domain to *partially* labeled target domain [15].

## 4.1 Datasets and Implementation Details

**Datasets.** We use two commonly used benchmark datasets (*i.e.*, Office-Home [60] and VisDA-2017 [46]) for SUDA and a large-scale dataset DomainNet [43] for MUDA and SSDA. 1) **Office-Home** [60] consists of images from four different domains: Art (Ar), Clipart (Cl), Product (Pr), and Real-World (Rw). Each domain contains 65 object categories in office and home environments. Following the typical settings [14, 12, 64, 40], we evaluate methods on one-source to one-target domain adaptation, resulting in 12 adaptation cases in total. 2) **VisDA-2017** [46] is a synthetic-to-real dataset for domain adaptation with over 280,000 images across 12 categories, where the source images are synthetic and the target images are real collected from MS COCO dataset [37]. 3) **DomainNet** [43] is a large-scale dataset containing about 600,000 images across 345 categories, which span 6 domains with large domain gap: Clipart (C), Infograph (I), Painting (P), Quickdraw (Q), Real (R), and Sketch (S). For MUDA, following the settings in [43, 66, 12, 33, 59], we evaluate methods on five-sources to one-target domain adaptation, resulting in 6 MUDA cases in total. For SSDA, we take the typical protocal in [22, 48, 12], where there are 7 SSDA cases conducted on the 4 sub-domains (*i.e.*, C, R, P and S) with 126 sub-categories selected from DomainNet. All methods are evaluated under the one-shot/three-shot setting respectively, where besides unlabeled samples, one/three sample(s) per class in the target domain are available during training.

**Implementation Details.** We apply our *ToAlign* on top of two different baseline schemes: *DANNP* [14, 64] and *HDA* [12]. *DANNP* is an improved variant of the classical adversarial learning based adaptation method DANN [17, 18], where the domain discrimination $D$ is conditioned on the predicted class probabilities. *HDA* is a state-of-the-art adversarial training based method which leverages the domain-specific representations as heuristics to obtain domain-invariant representations.

| Methods | Clipart | Infograph | Painting | Quickdraw | Real | Sketch | Avg. |
|---|---|---|---|---|---|---|---|
| Source-Only [21] | $47.6_{\pm 0.52}$ | $13.0_{\pm 0.41}$ | $38.1_{\pm 0.45}$ | $13.3_{\pm 0.39}$ | $51.9_{\pm 0.85}$ | $33.7_{\pm 0.54}$ | $32.9_{\pm 0.54}$ |
| ADDA(CVPR'17) [58] | $47.5_{\pm 0.76}$ | $11.4_{\pm 0.67}$ | $36.7_{\pm 0.53}$ | $14.7_{\pm 0.50}$ | $49.1_{\pm 0.82}$ | $33.5_{\pm 0.49}$ | $32.2_{\pm 0.63}$ |
| DANN(ICML'15) [17] | $45.5_{\pm 0.59}$ | $13.1_{\pm 0.72}$ | $37.0_{\pm 0.69}$ | $13.2_{\pm 0.77}$ | $48.9_{\pm 0.65}$ | $31.8_{\pm 0.62}$ | $32.6_{\pm 0.68}$ |
| DCTN(CVPR'18) [65] | $48.6_{\pm 0.73}$ | $23.5_{\pm 0.59}$ | $48.8_{\pm 0.63}$ | $7.2_{\pm 0.46}$ | $53.5_{\pm 0.56}$ | $47.3_{\pm 0.47}$ | $38.2_{\pm 0.57}$ |
| MCD(CVPR'18) [50] | $54.3_{\pm 0.64}$ | $22.1_{\pm 0.70}$ | $45.7_{\pm 0.63}$ | $7.6_{\pm 0.49}$ | $58.4_{\pm 0.65}$ | $43.5_{\pm 0.57}$ | $38.5_{\pm 0.61}$ |
| M$^3$SDA(ICCV'19) [43] | $57.2_{\pm 0.98}$ | $24.2_{\pm 1.21}$ | $51.6_{\pm 0.44}$ | $5.2_{\pm 0.45}$ | $61.6_{\pm 0.89}$ | $49.6_{\pm 0.56}$ | $41.5_{\pm 0.74}$ |
| M$^3$SDA-$\beta$(ICCV'19) [43] | $58.6_{\pm 0.53}$ | $26.0_{\pm 0.89}$ | $52.3_{\pm 0.55}$ | $6.3_{\pm 0.58}$ | $62.7_{\pm 0.51}$ | $49.5_{\pm 0.76}$ | $42.6_{\pm 0.64}$ |
| MDAN(NeurIPS'18) [70] | $60.3_{\pm 0.41}$ | $25.0_{\pm 0.43}$ | $50.3_{\pm 0.36}$ | $8.2_{\pm 1.92}$ | $61.5_{\pm 0.46}$ | $51.3_{\pm 0.58}$ | $42.8_{\pm 0.69}$ |
| MLMSDA(Arxiv'20) [36] | $61.4_{\pm 0.79}$ | $26.2_{\pm 0.41}$ | $51.9_{\pm 0.20}$ | $\mathbf{19.1}_{\pm 0.31}$ | $57.0_{\pm 1.04}$ | $50.3_{\pm 0.67}$ | $44.3_{\pm 0.57}$ |
| GVBG(CVPR'20) [14] | $61.5_{\pm 0.44}$ | $23.9_{\pm 0.71}$ | $54.2_{\pm 0.46}$ | $16.4_{\pm 0.57}$ | $67.8_{\pm 0.98}$ | $52.5_{\pm 0.62}$ | $46.0_{\pm 0.63}$ |
| CMSS(ECCV'20) [66] | $64.2_{\pm 0.18}$ | $\mathbf{28.0}_{\pm 0.20}$ | $53.6_{\pm 0.39}$ | $16.0_{\pm 0.12}$ | $63.4_{\pm 0.21}$ | $53.8_{\pm 0.35}$ | $46.5_{\pm 0.24}$ |
| HDA(NeurIPS'20) [12] | $63.6_{\pm 0.35}$ | $25.9_{\pm 0.16}$ | $56.1_{\pm 0.38}$ | $16.6_{\pm 0.54}$ | $69.1_{\pm 0.42}$ | $54.3_{\pm 0.26}$ | $47.6_{\pm 0.40}$ |
| Baseline | $66.4_{\pm 0.24}$ | $24.7_{\pm 0.16}$ | $57.3_{\pm 0.10}$ | $11.5_{\pm 0.17}$ | $69.2_{\pm 0.21}$ | $55.2_{\pm 0.13}$ | $47.3_{\pm 0.19}$ |
| Baseline+ToAlign | $\uparrow\mathbf{67.0}_{\pm 0.22}$ | $\uparrow 25.9_{\pm 0.20}$ | $\uparrow\mathbf{57.8}_{\pm 0.32}$ | $\uparrow 12.2_{\pm 0.14}$ | $\uparrow\mathbf{70.7}_{\pm 0.25}$ | $\uparrow\mathbf{56.0}_{\pm 0.18}$ | $\uparrow\mathbf{48.2}_{\pm 0.22}$ |

Table 2: Accuracy (%) of different MUDA methods on DomainNet with ResNet-101 as backbone. Best in bold.

| Method | | Acc. |
|---|---|---|
| DANNP | | 67.9 |
| DANNP+ToAlign | $s=1$ | 59.7 |
| | $s=8$ | 68.8 |
| | $s=16$ | 69.7 |
| | $s=64$ | 70.0 |
| | $s=128$ | 69.8 |
| | Adaptive $s$ | 69.9 |

Table 3: Ablation study on the influence of $s$ in Eq. 5.

| Method | Time/ms | GPU mem./MB | Acc./% |
|---|---|---|---|
| DANNP | 550 | 6,660 | 67.9 |
| DANNP+ MetaAlign[64] | 1,000 | 10,004 | 69.5 |
| DANNP+ ToAlign | 590 | 6,668 | 69.9 |

Table 4: Training complexity comparison (on GTX TITAN X GPU) in terms of computational time (of one iteration) and GPU memory for a mini-batch with batch size 32.

We use the ResNet-50 [21] pre-trained on ImageNet [30] as the backbone for SUDA, while using ResNet-101 and ResNet-34 for MUDA and SSDA respectively. Following [64, 40, 12], the image classifier $C$ is composed of one fully connected layer. The discriminator $D$ consists of three fully connected layers with inserted dropout and ReLU layers. We follow [69] to take an annealing strategy to set the learning rate $\eta$, *i.e.*, $\eta_t = \frac{\eta_0}{(1+\gamma p)^\tau}$, where $p$ indicates the progress of training that increases linearly from 0 to 1, $\gamma = 10$, and $\tau = 0.75$. The initial learning rate $\eta_0$ is set to $1e-3, 3e-4, 3e-4$, and $1e-3$ for SUDA on Office-Home, SUDA on VisDA-2017, MSDA on DomainNet, and SSDA on DomainNet, respectively. All reported experimental results are the average of three runs with different seeds.

## 4.2 Ablation Study

**Effectiveness of *ToAlign* on Different Baselines.** Our proposed *ToAlign* is generic and applicable to different domain adversarial training based baselines, where we focus on what features to align instead of the alignment methods. The last four rows in Table 1 show the ablation comparisons on Office-Home. Our *ToAlign* improves the accuracy of baseline *DANNP* and *HDA* by **2.0%** and **1.1%** respectively. As can be seen from the results in Table 1, Table 2, Table 5 and Table 6, our *ToAlign* can consistently bring significant improvement over the baseline schemes under different domain adaptation settings, *i.e.*, SUDA, MUDA and SSDA. *ToAlign* enables the domain alignment task to proactively serve the classification task, resulting in more effective feature alignment for image classification.

**Effectiveness of Different Ways to Obtain Positive Features.** As mentioned in Sec. 3.2, we use $\mathbf{w}_p^{cls} = s\mathbf{w}^{cls}$ as the attention weight (which conveys the classification prior/meta-knowledge) to derive positive feature $\mathbf{f}_p$, where $s$ is a parameter to modulate the energy of $\mathbf{f}_p$. We study the influence of $s$ under the setting of Rw$\rightarrow$Cl on Office-Home for our scheme *DANNP+ToAlign* and illustrate

| Methods | R→C | R→P | P→C | C→S | S→P | R→S | P→R | Avg. |
|---|---|---|---|---|---|---|---|---|
| Source-Only [21] | 55.6 | 60.6 | 56.8 | 50.8 | 56.0 | 46.3 | 71.8 | 56.9 |
| DANN(ICML'15) [17] | 58.2 | 61.4 | 56.3 | 52.8 | 57.4 | 52.2 | 70.3 | 58.4 |
| ADR(ICLR'18) [49] | 57.1 | 61.3 | 57.0 | 51.0 | 56.0 | 49.0 | 72.0 | 57.6 |
| CDAN(NeurIPS'18) [40] | 65.0 | 64.9 | 63.7 | 53.1 | 63.4 | 54.5 | 73.2 | 62.5 |
| ENT(NeurIPS'05) [20] | 65.2 | 65.9 | 65.4 | 54.6 | 59.7 | 52.1 | 75.0 | 62.6 |
| MME(ICCV'19) [48] | 70.0 | 67.7 | 69.0 | 56.3 | 64.8 | 61.0 | 76.1 | 66.4 |
| CANN(Arxiv'20) [47] | 72.7 | 70.3 | 69.8 | 60.5 | 66.4 | 62.7 | 77.3 | 68.5 |
| GVBG(CVPR'20) [14] | 70.8 | 65.9 | 71.1 | 62.4 | 65.1 | **67.1** | 76.8 | 68.4 |
| HDA(NeurIPS'20) [12] | 72.4 | 71.0 | 71.0 | **63.6** | 68.8 | 64.2 | 79.9 | 70.0 |
| HDA+ToAlign | **73.0**$_\uparrow$ | **72.0**$_\uparrow$ | **71.7**$_\uparrow$ | 63.0$_\downarrow$ | **69.3**$_\uparrow$ | 64.6$_\uparrow$ | **80.8**$_\uparrow$ | **70.6**$_\uparrow$ |

Table 5: Accuracy (%) of different one-shot SSDA methods on DomainNet with ResNet-34 as backbone. Best in bold.

| Methods | R→C | R→P | P→C | C→S | S→P | R→S | P→R | Avg. |
|---|---|---|---|---|---|---|---|---|
| Source-Only [21] | 60.0 | 62.2 | 59.4 | 55.0 | 59.5 | 50.1 | 73.9 | 60.0 |
| ADR(ICLR'18) [49] | 60.7 | 61.9 | 60.7 | 54.4 | 59.9 | 51.1 | 74.2 | 60.4 |
| CDAN(NeurIPS'18) [40] | 69.0 | 67.3 | 68.4 | 57.8 | 65.3 | 59.0 | 78.5 | 66.5 |
| ENT(NeurIPS'05) [20] | 71.0 | 69.2 | 71.1 | 60.0 | 62.1 | 61.1 | 78.6 | 67.6 |
| MME(ICCV'19) [48] | 72.2 | 69.7 | 71.7 | 61.8 | 66.8 | 61.9 | 78.5 | 68.9 |
| MetaMME(ECCV'20) [33] | 73.5 | 70.3 | 72.8 | 62.8 | 68.0 | 63.8 | 79.2 | 70.1 |
| GVBG(CVPR'20) [14] | 73.3 | 68.7 | 72.9 | 65.3 | 66.6 | **68.5** | 79.2 | 70.6 |
| CANN(Arxiv'20) [47] | 75.4 | 71.5 | 73.2 | 64.1 | 69.4 | 64.2 | 80.8 | 71.2 |
| HDA(NeurIPS'20) [12] | 74.5 | 71.5 | 73.9 | 65.9 | 70.1 | 65.9 | 81.9 | 71.8 |
| HDA+ToAlign | **75.7**$_\uparrow$ | **72.9**$_\uparrow$ | **75.6**$_\uparrow$ | **66.2**$_\uparrow$ | **71.1**$_\uparrow$ | 66.4$_\uparrow$ | **83.0**$_\uparrow$ | **73.0**$_\uparrow$ |

Table 6: Accuracy (%) of different three-shot SSDA methods on DomainNet with ResNet-34 as backbone. Best in bold.

the results in Table 3. As discussed around Eq. (5), we can use an adaptively calculated $s$, which achieves 2% improvement over the baseline on target test data. Moreover, we can treat $s$ as a preset hyper-parameter. We found that the performance drops drastically if $s$ is too small (*e.g.*, $s = 1$). That is because the energy of the source positive feature will get too weak when $s$ gets too small (*e.g.*, the source feature $\mathbf{f}$'s average energy $\mathcal{E}(\mathbf{f})$ is about 800; if $s = 1$, the source positive feature's average energy $\mathcal{E}(\mathbf{f}_p)$ is about 2). Then, it would be ineffective to align the target with the source positive features. When $s$ is larger than 16, the performance significantly outperforms the baseline and approaches the result of using adaptive $s$. As an optional design choice, we could transform the weight $\mathbf{w}^{cls}$ with certain activation function $\sigma(\cdot)$ such as Sigmoid or Softmax followed by a best selected scaling factor $s$, *i.e.*, $\mathbf{w}_p^{cls} = s\sigma(\mathbf{w}^{cls})$. We found the results (*i.e.*, 69.6/69.7 for Sigmoid/Softmax) are close to that without activation function. We reckon that what is more important is the relative importance among the elements in $\mathbf{w}^{cls}$. For simplicity, we finally take the adaptive $s$ (cf. Eq. 5) for all experiments.

### 4.3 Comparison with the State-of-the-arts

**Single Source Unsupervised Domain Adaptation (SUDA).** We incorporate our *ToAlign* into the recent state-of-the-art UDA method *HDA* [12], denoted as *HDA+ToAlign*. Table 1 shows the comparisons with the previous state-of-the-art methods on Office-Home. *HDA+ToAlign* outperforms all the previous methods and achieves the state-of-the-art performance. It is noteworthy that *HDA+ToAlign* achieves the best adaptation results on almost all the one-source to one-target adaptation cases thanks to the effective feature alignment for classification. The results on VisDA-2017 could be found in Appendix, where *HDA+ToAlign* outperforms *HDA* by 0.9%.

**Multi-source Unsupervised Domain Adaptation (MUDA).** Table 2 shows the results on Domain-Net, where all the methods take ResNet-101 as the feature extractor. We build our *Baseline* based on *HDA* [12]. For simplicity, we replace the multi-class domain discriminator in the original *HDA* by a two-class one as in [58, 17, 66]. Note that CMSS [66] selects suitable source samples for alignment while our *ToAlign* selects task-discriminative sub-feature for each sample for task-oriented alignment. Compared with *Baseline*, *ToAlign* brings about 0.9% improvement and helps to achieve the best performance on this more challenging dataset.

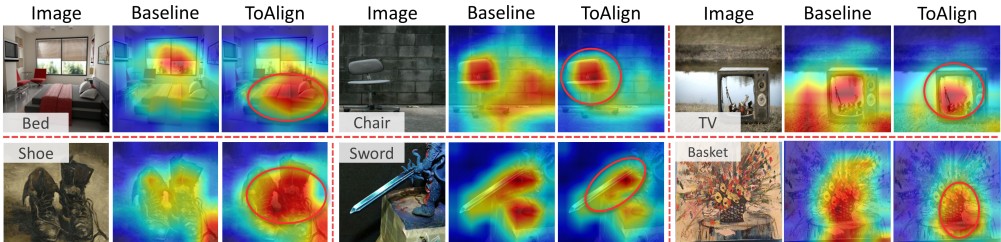

Figure 5: Visualization of the feature response maps on target test images. First row: Art of Office-Home. Second row: Painting of DomainNet.

**Semi-supervised Domain Adaptation (SSDA).** Table 5 and Table 6 show the results on one-shot and three-shot SSDA respectively, where all the methods use ResNet-34 as backbone. To compare with previous methods, we apply *ToAlign* on top of *HDA*. *HDA+ToAlign* outperforms *HDA* by 0.6%/1.2% for one-/three-shot settings, and surpasses all previous SSDA methods.

### 4.4 Complexity

In Table 4, we compare the training complexity and performance of *ToAlign* with baseline *DANNP*, and *DANNP+MetaAlign* [64] which incorporates meta-learning to coordinate the optimization of domain alignment and image classification. In contrast, inspired by the prior knowledge of what feature should be aligned to serve classification task, we distill such meta-knowledge from classification task and explicitly pass it to alignment task for classification-oriented alignment, eschewing complex optimization. Compared with baseline, *ToAlign* introduces negligible additional computational cost (only 7%) and occupies almost the same GPU memory as the baseline, which is much smaller than that of *DANNP+MetaAlign*, which almost doubles the computational cost due to its complex meta-optimization. Thanks to our explicit design which makes domain alignment effectively serve the classification task, our *ToAlign* achieves superior performance to *MetaAlign*.

### 4.5 Feature Visualization

We visualize the target feature response maps $F$ (which will be pooled to be the input of the image classifier) of the *Baseline* (DANNP) and *ToAlign* in Figure 5. *Baseline* sometimes focuses on the background features which are useless to the image classification task, since it aligns the holistic features without considering the discriminativeness of different channels/sub-features. Thanks to our task-oriented alignment, in *ToAlign*, the features with higher responses are in general related to task-discriminative features, which is more consistent with human perception. More results can be found in the Appendix.

## 5 Conclusion

In this paper, we study what features should be aligned across domains for more effective unsupervised domain adaptive image classification. To make the domain alignment task proactively serve the classification task, we propose an effective task-oriented alignment (*ToAlign*). We explicitly decompose a feature in the source domain into a task-related feature that should be aligned and a task-irrelevant one that should be ignored, under the guidance of the meta-knowledge induced from the classification task itself. Extensive experiments on various datasets demonstrate the effectiveness of our *ToAlign*. In our future work, we will extend *ToAlign* to tasks beyond image classification, *e.g.*, object detection and segmentation.

## Acknowledgments and Disclosure of Funding

This work was supported in part by the National Key Research and Development Program of China 2018AAA0101400 and NSFC under Grant U1908209, 61632001, and 62021001.

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
