# ToAlign: Task-oriented Alignment for Unsupervised Domain Adaptation (Appendix)

**Guoqiang Wei**[1*] **Cuiling Lan**[2†] **Wenjun Zeng**[2] **Zhizheng Zhang**[2] **Zhibo Chen**[1†]

[1] University of Science and Technology of China    [2] Microsoft Research Asia
wgq7441@mail.ustc.edu.cn   {culan,wezeng,zhizzhang}@microsoft.com
chenzhibo@ustc.edu.cn

## A. Visualization of Decomposed Features

To better understand and validate the discriminativeness of the positive and the negative features, similar to Figure 4 in our manuscript, here we show more visualization results of the spatial maps $F$ with channels modulated by $\mathbf{w}^{cls}$ (corresponding to positive features) and $-\mathbf{w}^{cls}$ (corresponding to negative features) following [10, 15]. We can observe that the positive information is more related to the foreground objects that provide the discriminative information for the classification task, while the negative one is more in connection with the non-discriminative background regions.

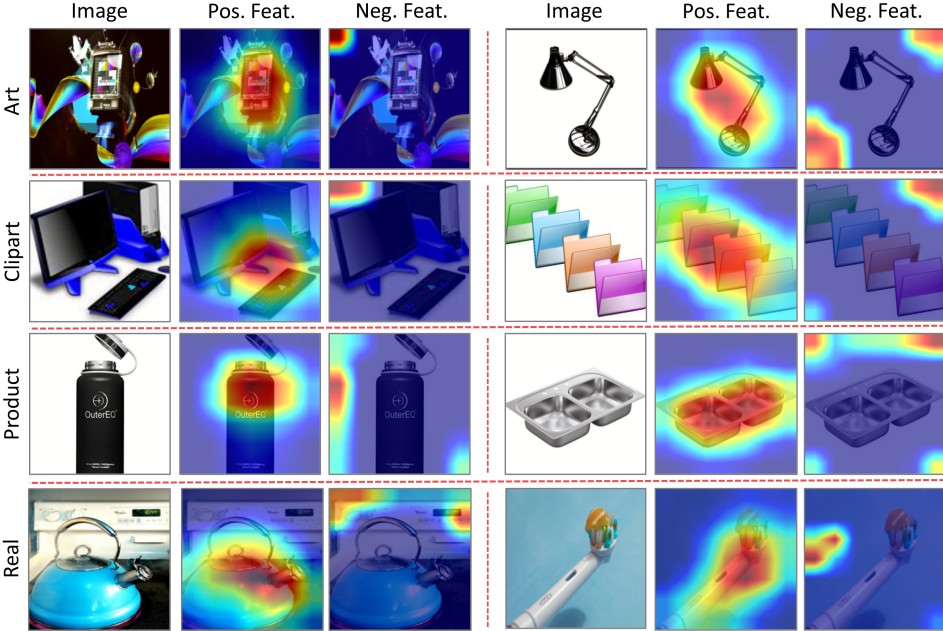

Figure 1: Visualization of task-discriminative and task-irrelevant features. The images are sampled from different domains of Office-Home.

---

*This work was done when Guoqiang Wei was an intern at MSRA.
†Corresponding author.

35th Conference on Neural Information Processing Systems (NeurIPS 2021).

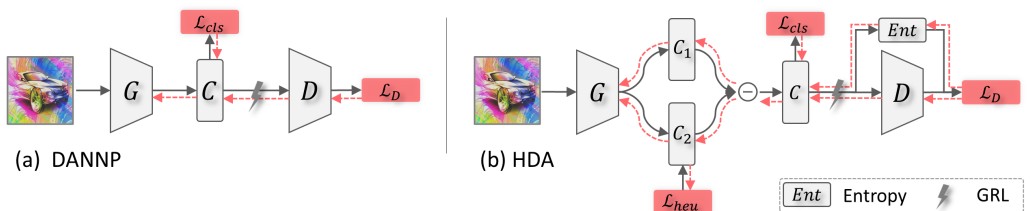

Figure 2: Pipelines of two representative domain alignment based UDA methods. (a) DANNP [12]. (b) HDA [2]. $\mathcal{L}_{heu}$ denotes a heuristic loss which is implemented by $\mathcal{L}_1-$norm loss [2].

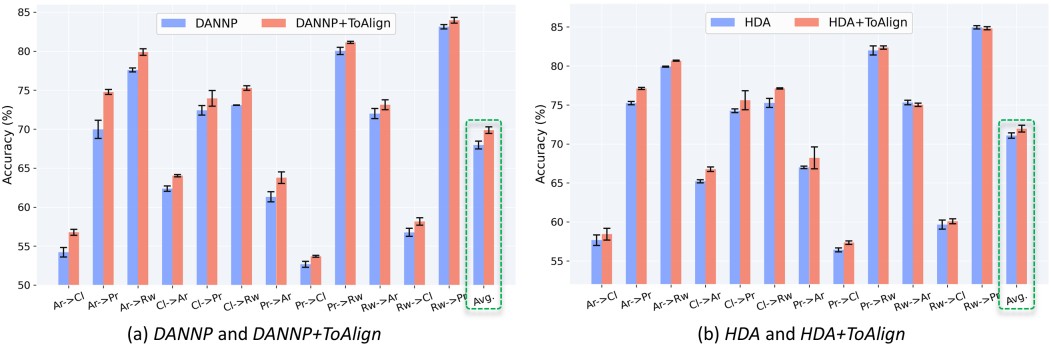

Figure 3: Error bars of *ToAlign* on top of *DANNP* and *HDA* on Office-Home.

## B. Experiments

### B.1 More Implementation Details

We use two domain alignment based methods as our baselines: 1) **DANNP** [12] is an improved variant of DANN [4], where the domain discrimination $D$ in DANNP is conditioned on the predicted class probabilities instead of extracted features as illustrated in Figure 2 (a). 2) **HDA** [2] draws inspiration from heuristic search and incorporates the domain-specific representations as heuristics to help learn domain-invariant ones. Figure 2 (b) shows its architecture.

All experimental results are obtained by running three times with different seeds. To evaluate the stableness of our *ToAlign*, we visualize error bars of our schemes *DANNP+ToAlign* and *HDA+ToAlign* on Office-Home in Figure 3, where we also present the error bars for the two baseline schemes *DANNP* and *HDA*. The variances between our *ToAlign* and the corresponding baselines are close (0.41 vs. 0.40 for DANNP and 0.40 vs. 0.35 for HDA) and our *ToAlign* dose not introduce much additional unstability.

### B.2 Experimental Results of SUDA

As referred to in our main manuscript, the experimental results on Visda-2017 for SUDA are presented in Appendix. Here, Table 1 shows the results, where our *ToAlign* introduces 0.9% improvements over the baseline *HDA*.

### B.3 Feature Visualization

We visualize more results of the feature response maps on the target test images in Figure 4, as a supplement to Figure 5 in our main manuscript. *Baseline* sometimes focuses on the background features which are useless to the image classification task, since it aligns the holistic features without considering the discriminativeness of different channels/sub-features. Thanks to our task-oriented alignment, in *ToAlign*, the features with higher responses are in general related to task-discriminative features, which is more consistent with human perception.

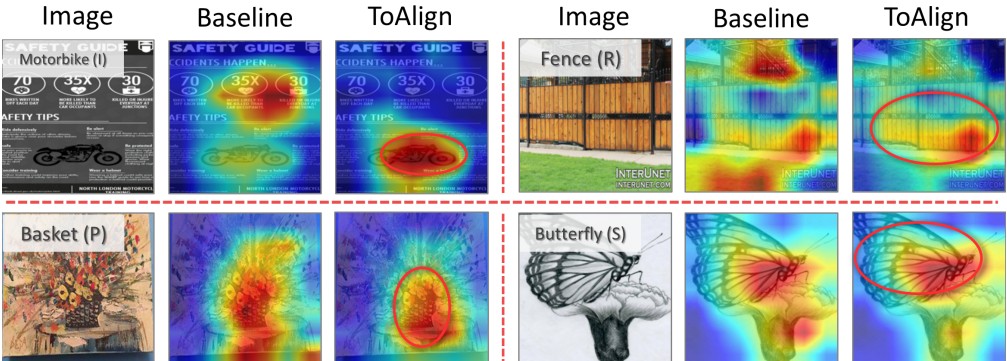

Figure 4: Visualization of the feature response maps on target test images. The Category (Domain) information is shown on each sample.

| Method | Avg. |
|---|---|
| Source-Only [5] | 55.3 |
| DANN(ICML'15) [4] | 57.4 |
| CDAN(NeurIPS'18) [7] | 70.0 |
| MDD(ICML'19) [14] | 74.6 |
| GVB(CVPR'20) [3] | 75.3 |
| HDA(NeurIPS'20) [2] | 74.6 |
| HDA+ToAlign | 75.5 |

Table 1: Classification accuracy (%) of the Synthetic → Real setting on Visda-2017 for SUDA using ResNet-50 as backbone. Note that HDA [2] does not report the result on this dataset and we obtain the result by running their released source code.

We further visualize the learned source (red) and target (blue) feature representations (*i.e.*, $\mathbf{f}_s$ and $\mathbf{f}_t$) using t-SNE [9] for different methods in Figure 5. Figure 5 (a) shows the embedded features of the Source-Only method where no adaptation technique is used, where we can see that the samples are very scattered. In comparison, the samples for *HDA* [2] (cf. Figure 5 (b)) and our *HDA+ToAlign* (cf. Figure 5 (c)) form more compact clusters, where the clusters of ours are more compact and the target samples are located closer to the source samples than *HDA*.

## C. Broader Impact

Unsupervised domain adaptation aims to obtain better performance on unlabeled target data based on the knowledge from labeled source data and unlabeled target data, which is an important and practical problem in both the academic and industry. Our proposed *ToAlign* emphasizes that domain alignment task should assist/serve classification task, where we perform alignment under the guidance of the meta-knowledge induced from classification task. We also provide some understanding from the meta-knowledge perspective, where we pass the meta-train task knowledge in a simple and effective way to the meta-test task. This provides some insights on how to pass meta-knowledge more effectively for the meta-learning based multi-task communication [6, 12, 1].

The major societal impact of our *ToAlign* arises from the UDA task itself, which aims to transfer knowledge from labeled source domain to unlabeled target domain, leading to heavy dependency on source domain. The major limitation of our *ToAlign* is that it is only applicable to domain adversarial learning based UDAs, which though dominates in the top performance methods. How to apply the idea to other category of methods, *e.g.*, pseudo-label based ones [8, 13, 11], will be investigated in future.

*ToAlign* could be further improved from two perspectives. First, other ways to derive classification meta-knowledge could be exploited, where we now use the gradients as guidance which is drawn inspiration from Grad-CAM [10]. Second, *ToAlign* could be further expanded to more challenging tasks like semantic segmentation and object detection.

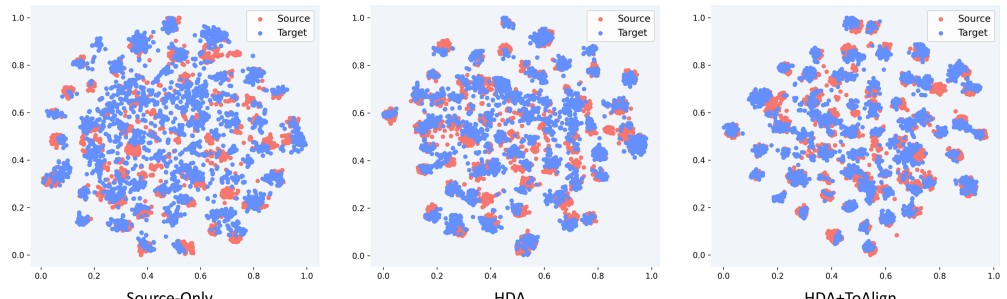

Figure 5: T-sne visualization of different methods on Ar→Pr of Office-Home. Red: source. Blue: target.