# OpenReview forum: "ToAlign: Task-Oriented Alignment for Unsupervised Domain Adaptation"
_NeurIPS.cc/2021/Conference — NeurIPS 2021 Poster_

### Official Review · Reviewer_qjmz · 2021-07-11

**Rating:** 7
**Confidence:** 4

**Summary:**

The paper proposed a method "To-Align" for alignment of the domains in subspaces that have relevant information embedded for a given task. For the classification task, the paper demonstrates that using Grad-CAM for obtaining subspaces is effective. Results on the single source and multi-source benchmarks show consistent improvements in the performance of UDA.

The motivation behind the usage of attention for domain adaptation has been well studied in the existing literature. To fully understand the benefits of the proposed method, more experimental studies would be necessary. However, in the current form, the paper is marginally acceptable.

**Limitations And Societal Impact:**

Limitations:

* The authors point out that this approach works only with in domain adversarial settings and not in a pseudo-labelling settings. The * authors are encouraged to inlude limitations w.r.t to the usage of Grad-CAM in the feature selection process and how it may/may not improve other baselines.

Societal Impact:
* The authors should comment on any negative bias carried over by the Grad-CAM based feature selection.
* Does "ToAlign" ensure fairness if the source dataset consists of imbalanced classes?
* Would "ToAlign" carry over source dataset bias in medical applications such as tumor segmentation?

**Main Review:**

The paper is of good quality and clarity. However, it is not up to the mark with regard to originality and significance (idea & results). Please see the below full review below.

1. Using attention for DA has already been explored in many papers [Ref 1,2]. The goal of this line of research is to identify the interesting aspects in the image for the given task. There have also been several methods that consider feature level manipulation (such as negative and positive features) for DA [Ref 3]. It is not clear why the proposed "ToAlign" performs better than any other attention-based works as the paper does not compare against them.

2. Recent work on MUDA at NeurIPS [Ref 4] showed remarkable performance on DomainNet while being simpler than existing MUDA methods. Please comment on the performance of your method against this work.

3. The experimental setup: The current set of experiments demonstrate improvement over a couple of baselines. How would the proposed "ToAlign" framework help with respect to other methods? Can any method exploit the benefits of "ToAlign"? What are the limitations? I would expect that 'ToAlign' will not improve over [Ref 1,2].

4. The method demonstrates only a marginal overall improvement (0.7%) on the SSDA dataset. However, the method is consistently improving the baseline as shown in Table  1 and Table 2.

5. Are authors planning to release a reference implementation?

6. Minor comments
 - L193:: Classifier C and #channels C have the same notation.

-----
Please cite these related work and mention how these methods are different from yours:-

[Ref 1] Vinod Kumar Kurmi, Shanu Kumar, Vinay P. Namboodiri, "Attending to Discriminative Certainty for Domain Adaptation," CVPR'19

[Ref 2] Guoliang Kang, Liang Zheng, Yan Yan, and Yi Yang, "Deep Adversarial Attention Alignment for Unsupervised Domain Adaptation: the Benefit of Target Expectation Maximization", ECCV'18

[Ref 3] Jogendra Nath Kundu, Naveen Venkat, Ambareesh Revanur, Rahul M V, R. Venkatesh Babu, "Towards Inheritable Models for Open-Set Domain Adaptation" CVPR'20

[Ref 4] Naveen Venkat, Jogendra Nath Kundu, Durgesh Kumar Singh, Ambareesh Revanur, Venkatesh Babu R., "Your Classifier can Secretly Suffice Multi-Source Domain Adaptation", NeurIPS'20

**Time Spent Reviewing:**

20

---

> ### Author Response · Authors · 2021-08-10
> **Response to Reviewer qjmz (Q7 to Q8)**
>
> **Q7: Limitations: (1) The authors are encouraged to include limitations w.r.t to the usage of Grad-CAM in the feature selection process and how it (2)may/(3)may not improve other baselines.**
>
> **A7:** Thank you for the valuable suggestion, and we will add the discussion of the limitations w.r.t. the usage of Grad-CAM in the revision.
>
> 1. The attention weights obtained by Grad-CAM may be imperfect, since it is still possible for the Grad-CAM to be affected by noise in some case [ref 7].
> 2. The key to the improvement over baselines is that we select the “right” (classification-discriminative) source features to be aligned with target domain.
> 3. However, ToAlign may not be applicable to UDAs with complex alignment designs, e.g., the alignment procedure only involves target domain without source domain in MCD [43] and STAR [37].
>
> References:
>
> > [ref 7] Zhang, Qinglong, Lu Rao, and Yubin Yang. "Group-CAM: Group Score-Weighted Visual Explanations for Deep Convolutional Networks." arXiv preprint arXiv:2103.13859 (2021).
>
> **Q8: Societal Impact: (1) The authors should comment on any negative bias carried over by the Grad-CAM based feature selection. (2) Does "ToAlign" ensure fairness if the source dataset consists of imbalanced classes? (3) Would "ToAlign" carry over source dataset bias in medical applications such as tumor segmentation?**
>
> **A8:** Thank you for this valuable suggestion, and we will discuss them in the Broader Impact section in the revision.
>
> The major societal impact of our ToAlign arises from the UDA task itself, which aims to transfer knowledge from labeled source domain to unlabeled target domain, leading to heavy dependance on source domain.
>
> 1. The Grad-CAM is operated on the source domain, where the fully supervised learning makes sure that it works in most cases. However, it may be biased towards wrong classification-discriminative features if the baseline model has too poor performance on source domain
> 2. The fairness under imbalanced classes settings is determined by the performance of Grad-CAM. However, previous works [ref 8] demonstrated that Grad-CAM is applicable to classes imbalanced dataset since the ground-truth category label is used for Grad-CAM.
> 3. Yes, because only the source domain is annotated in UDA. However, bias correction technologies [ref 9] can be exploited to alleviate medical dataset bias.
>
> References:
>
> > [ref 8] Trong, Vo Hoang, Yu GwangHyun, and Kim JinYoung. "Yielding Multi-Fold Training Strategy for Image Classification of Imbalanced Weeds." Applied Sciences, 2021.

---

> > ### Comment · Reviewer_qjmz · 2021-08-17
> > **Discussion**
> >
> > Yes. Thank you for the response. As machine learning becomes more pervasive in our society, it is important for others to understand the limitations of proposed methods and negative impact on society.

---

> ### Author Response · Authors · 2021-08-10
> **Response to Reviewer qjmz (Q1 to Q6)**
>
> We thank you for the positive comments on the quality and clarity, and constructive suggestions on improving this work. Please find our responses below:
>
> **Q1: Attention/feature manipulation for DA has already been explored in many papers [Ref 1, 2, 3]. It is not clear why the proposed "ToAlign" performs better than any other attention-based works as the paper does not compare against them.**
>
> **A1:** Thanks for this comment and suggestion, we will include the suggested papers for comparison in the revision. Note that we have analyzed the differences from other attention-based methods [21, 23, 25, 54] in Sections 1 and Section 2. The reason our ToAlign performs better than others is that *we explicitly introduce task meta-knowledge (class-specific responses by Grad-CAM using the ground-truth labels on source domain) into the alignment process to achieve task-oriented alignment*. However, the weights in other attention-based solutions are commonly heuristically learned and it is possible for them to focus on class-irrelevant features which will hinder efficient alignment (as we will discuss below). We provide detailed comparison analysis for the suggested papers as below:
>
> **[ref 1]** incorporates the certainty of the discriminator while training the classifier, which is similar to [54] that we have discussed in section 1. Their attention is selected **based on the domain alignment task**. As claimed in [56], such self-induced attention will focus on classification-irrelevant regions, leading to inefficient alignment/adaptation. ToAlign differs from [ref 1, 54] in:
> 1. our attention is obtained **from the perspective of classification task instead of domain alignment task**,
> 2. our attention is derived from the gradients using ground-truth source labels instead of the uncertainty of domain discriminator prediction.
>
> To compare with such domain-derived attention methods, we build a SE [ref 5] model based on the strong baseline DANNP, where the attention weights in SE block is updated/learned only by the domain alignment task and used for classification task directly. The SE block is inserted in the final layer of the backbone, same as ToAlign. The domain-derived attention improves DANNP (67.9% on Office-Home) by a slim margin (0.1%), where such attention may focus on task-irrelevant features without guidance of classification task. As a comparison, ToAlign can improve DANNP by 2.0% without any additional parameters.
>
> **[ref 2]** is an **image-translation based** method, another category of UDAs, with a consistency regularization on the attention maps for paired inputs. As shown in Figure 5 (main paper) and Figure 4 (appendix), such attention maps obtained directly from the backbone’s outputs may focus on classification-irrelevant regions without the direct guidance of classification task. Note that [ref 2] requires additional CycleGAN models to generate synthetic images.  In contrast, our proposed ToAlign belongs to domain-alignment based UDAs and derives the attention based on the gradients of the final prediction of ground-truth category without additional models, thus attention is highly related to the classification task.
>
> **[ref 3]**: The positive and negative features in [ref 3] are corresponding to **source-domain features** and **out-of-source-distribution (OOD) features** respectively, which are then used to help measure domain shift for open-set DA. ToAlign fundamentally differs from [ref 3] in:
> 1. Our positive/negative features are derived **from the perspective of classification task, instead of domain**.
> 2. Our positive source features are used to align with target domain directly, instead of aiming to help measure domain shift.
> 3. Our feature decomposition is implemented based on the simple Grad-CAM, while [ref 3] requires feature-splicing and clustering on negative features.
>
> References:
>
> >[ref 5] Hu, Jie, Li Shen, and Gang Sun. "Squeeze-and-excitation networks." In CVPR. 2018.
>
> **Q2: Recent work on MUDA at NeurIPS [Ref 4] showed remarkable performance on DomainNet while being simpler than existing MUDA methods. Please comment on the performance of your method against this work.**
>
> **A2:** Our ToAlign differs from SImpAl [ref 4] in:
> 1. SImpAl is based on pseudo-labeled target samples and its performance heavily depends on the selection of pseudo-labels (section 4.2(e) in SImpAl), while ToAlign is based on domain alignment,
> 2. SImpAl only evaluates under the multi-source UDA settings, while we evaluate the effectiveness on single-source UDA, multi-source UDA and semi-supervised DA.
>
> | Method | Clipart | Infograph | Painting | Quickdraw | Real | Sketch | Avg. |
> | :-: | :-: | :-: | :-: | :-: | :-: | :-: | :-: |
> | M3SDA | $57.2_{\pm0.9}$ | $24.2_{\pm1.2}$ | $51.6_{\pm0.4}$ | $5.2_{\pm0.4}$ | $61.6_{\pm0.9}$ | $49.6_{\pm0.5}$ | $41.5_{\pm0.7}$ |
> | SImpAl | $66.4_{\pm0.8}$ | $\mathbf{26.5}_{\pm0.5}$ | $56.6_{\pm0.7}$ | $\mathbf{18.9}_{\pm0.8}$ | $68.0_{\pm0.5}$ | $55.5_{\pm0.3}$ | $48.6_{\pm0.6}$ |
> | ToAlign | $\mathbf{67.0}_{\pm0.2}$ | $25.9_{\pm0.2}$ | $\mathbf{57.8}_{\pm0.3}$ | $12.2_{\pm0.1}$ | $\mathbf{70.7}_{\pm0.2}$ | $\mathbf{56.0}_{\pm0.2}$ | $48.2_{\pm0.2}$ |
>
> Our proposed ToAlign outperforms SImpAl on most domains except for the Infograph and Quickdraw. For Infograph, we achieve competitive performance as SImpAl. For Quickdraw, ToAlign is anomalously inferior to SImpAl, because the Quickdraw is much more noisy than other domains [41, ref 6]. Please note that this domain has been suggested to be discarded for evaluation [41, ref 6]. If we follow this setting to discard Quickdraw for more reliable evaluation, as shown in the table below, ToAlign outperforms SImpAl by 0.9% statistically (Avg.).
>
> | Method | Clipart | Infograph | Painting | Quickdraw | Real | Sketch | Avg. |
> | :-: | :-: | :-: | :-: | :-: | :-: | :-: | :-: |
> | SImpAl* | 66.4 | **26.5** | 56.6 |-- | 68.0 | 55.5 | 54.6 |
> | ToAlign* | **67.0** | 25.9 | **57.8** | -- | **70.7** | **56.0** | 55.5 |
>
> References:
>
> > [ref 6] Zhou, K., Yang, Y., Qiao, Y. and Xiang, T., 2020. Domain adaptive ensemble learning. arXiv preprint arXiv:2003.07325.
>
> **Q3: How would the proposed "ToAlign" framework help with respect to other methods?**
>
> **A3:** Our proposed ToAlign is applicable to most alignment-based UDA solutions by introducing task guidance to achieve task-orientated alignment. As for the limitation, as clarified in the section of Broader Impact in our appendix, applying the idea to other category of methods, e.g., pseudo-labeled based ones and image-translation based ones, has not been studied in this work and is worth being investigated in the future work.
>
> ToAlign is complementary to [ref 1] which introduces guidance signals from alignment to help classification, while ours introduces guidance signals from classification to help alignment. Thus, they can be easily combined. [ref 2] is based on image-translation with a consistency regularization on paired inputs, where the (target, synthetic source) pair is unlabeled. However, the Grad-CAM used in ToAlign requires annotations on source domain. Therefore, ToAlign is not applicable to [ref 2].
>
> **Q4: The method demonstrates only a marginal overall improvement (0.7%) on the SSDA dataset. However, the method is consistently improving the baseline as shown in Table 1 and Table 2.**
>
> **A4:** For SSDA, there are two commonly used settings, i.e., one-shot and three-shot. We improve the state-of-the-art baseline by 1.2% under the three-shot setting in Table 3 (main paper), and 0.6% (in Table 2 of the appendix) under the one-shot setting. The one-shot setting is much more challenging considering that there are only 1/3 target samples are labeled compared with three-shot setting.
>
> **Q5: Are authors planning to release a reference implementation?**
>
> **A5:** We will release our implementation upon the paper acceptance.
>
> **Q6: Minor comments**
>
> **A6:** Thank you for your nice suggestion, we will modify the annotations for easier reading in the revision.

---

> > ### Comment · Reviewer_qjmz · 2021-08-17
> > **Discussion**
> >
> > I am very happy with the response you provided and find your response convincing.
> >
> > Disc-A1. Thanks for providing detailed response and commenting how "ToAlign" differs from the other attention based papers.
> > I agree that the features and subspaces are selected from classification point of view rather than from a domain perspective.
> >
> > [ref 1] Thanks for pointing out the similarities of DANNP and [ref 1]. The results clearly indicate that ToAlign outperforms the domain based alignment (2% v/s 0.1% improvements).
> >
> > [ref 2,3] I agree that translation-based/feature-splicing based maybe not be as effective as GradCAM based feature selection. Please note that [ref 3] is a source-free method and is therefore tackling a more challenging problem. I find the idea presented in [ref 3] simple -- it does not use gradient for positive-negative subspaces. ToAlign is also simple since it utilized the well-studied GradCAM method. Thanks again for eloborating and clarifying on the question Q1.
> >
> > Disc-A2. I agree that ToAlign is an impressive idea that works well across single source and multi source domain adaptation problems. I appreciate the authors for providing analysis. I would not expect a method to obtain state-of-the-art across all splits in DomainNet.
> >
> > - (a) I am not aware of any noise in Quickdraw dataset. Thanks for pointing out a reference paper.
> > - (b)  For Table. 2, it seems like you did not exclude "Quickdraw" from training. Maybe performance of your approach would improve further, if you did. I am convinced that ToAlign is an impressive approach.
> >
> > Disc-A3. I agree that it is interesting to include GradCAM based feature selection on both branches - alignment and classification. Yes, there is scope for further research in this avenue.
> >
> > Disc-A4. I understand the challenges with one-shot settings
> >
> > Disc-A5,A6. Thanks for the response and citing the additional related works indicated in the review.

---

> > > ### Author Response · Authors · 2021-08-20
> > > **Response to Discussion**
> > >
> > > We sincerely thank you for the positive comments on our responses.
> > >
> > > **Answer to Disc-A2 (b): For Table. 2, it seems like you did not exclude "Quickdraw" from training. Maybe performance of your approach would improve further, if you did. I am convinced that ToAlign is an impressive approach.**
> > >
> > > We further conduct an experiment where the *Quickdraw* domain is completely discarded from the source domains during training. Compared with *ToAlign*$^{*}$ in Table 2, the performance is improved from $55.5_{\pm0.236}$ to $55.8_{\pm0.112}$. The slight decrease in STD (i.e., from 0.236 to 0.112) indicates discarding this noisy domain promotes learning stableness.
> > >
> > > This noisy domain not only cannot be taken as a reliable evaluation data, nor can it contribute much to training. Thanks for your insightful suggestion. We will uncover this to our community in our revision.

---

> > > > ### Comment · Reviewer_qjmz · 2021-08-25
> > > > **Thank you.**
> > > >
> > > > Thanks for the update! Well, the results seem to show an insignificant improvement making it a weak case for discarding Quickdraw altogether. Nevertheless, I appreciate the authors for the results and showing the strengths of ToAlign method.

---

### Official Review · Reviewer_9jPd · 2021-07-16

**Rating:** 6
**Confidence:** 4

**Summary:**

The authors propose ToAlign, a method which reweights the source features in an adversarial domain adaptation pipeline based on their relevance in predicting the source label. This method could be used in conjunction with any alignment-based domain adaptation approach and has shown the ability to improve performance by focusing alignment on relevant features. The authors show the results of experiments in multiple domain adaptation settings and consistently improve upon baseline performance.

**Limitations And Societal Impact:**

The authors don't discussion potential negative societal impacts of their work.

**Main Review:**

The main ideas of the work are clearly presented; using ToAlign to reweight source features before aligning source and target distributions is shown to empirically improve performance on several actively explored domain adaptation tasks. The relevance of this proposed approach is demonstrated not just by its performance on challenging benchmarks, but also due to the fact that it can be easily combined with other state-of-the-art domain adaptation approaches to further improve performance.

While the underlying math and experiments are clearly defined and discussed, elements of the motivation of this work are confusing or don't contribute to a better understanding of the work. Particularly the understanding from the meta-knowledge perspective, detracts from clearly understanding the work's motivations and intuitions. Is the meta-knowledge discussed in this section shared or accumulated across multiple examples or is instance-specific? If instance-specific is meta-knowledge the appropriate mechanism of conceptualizing this information?

While to the best of my knowledge the approach proposed in this work is original, the treatment of prior work leaves much to be desired. The work presents its as one of the only approaches to focus on task oriented alignment, but ignores several works on disentangled representations for domain adaptation which attempt the same goal. Positioning the works improvements and motivations solely on the focusing on task-oriented features and ignoring how it relates to other work which attempt the same goal may lead to inaccurate takeaways from the paper and is a substantial concern with the quality of the work.
(Domain Agnostic Learning with Disentangled Representation - Peng et al)
(Learning Disentangled Semantic Representation for Domain Adaptation - Cai et al)
(DiDA: Disentangled Synthesis for Domain Adaptation - Cao et al)

**Time Spent Reviewing:**

7

---

> ### Author Response · Authors · 2021-08-10
> **Response to Reviewer 9jPd**
>
> We thank you for the positive comments on the originality of our approach and paper writing, valuable suggestions on improving the understanding of this work. Please find our responses below:
>
> **Q1: Is the meta-knowledge discussed in this section shared or accumulated across multiple examples or is instance-specific? If instance-specific is meta-knowledge the appropriate mechanism of conceptualizing this information?**
>
> **A1:**  It is shared across different examples of the same class. We are sorry for this confusion and will clarify it in the revision.  In our proposed method, the meta-knowledge aims to uncover how to filter the task-related (classification) feature from the holistic feature extracted by the backbone, which is thus reflected on the attention weights $\mathbf{w_p^{cls}}$. We obtain the attention weights using the gradients of the final prediction of the ground-truth category (Grad-CAM), and observe that the attention weights become consistent within each class after the model converges. Therefore, although the task-oriented feature $\mathbf{f_p^s}$ is instance-specific, the meta-knowledge lies in the class level and is shared within each class.
>
> **Q2: While to the best of my knowledge the approach proposed in this work is original, the treatment of prior work leaves much to be desired.**
>
> **A2:**  Thanks for your valuable suggestion. We have clarified the difference from disentanglement-based [23, 25] in Section 2, and we will add more detailed discussion as suggested in our revision. Despite the similar goal, our proposed method is fundamentally different from other disentanglement-based works in terms of the **principle and methodology** of information decomposition.
>
> Specifically, [ref 1, 23] learn domain-invariant/common features via disentanglement from the perspective of domain labels rather than from the perspective of task (classification) relevance like ours.
>
> [ref 2, ref 3] perform the disentanglement from the perspective of semantics and domain simultaneously and only adopt domain-invariant semantics for inference after the disentanglement, leaving domain-specific but task-related information underexplored (i.e., resulting in an information loss during the disentanglement process). It is noteworthy that [ref 2, ref 3] implement disentanglement with the VAE framework which entails additional decoder(s) and complex losses design (e.g., ELOB, reconstruction and orthogonal losses).  In contrast:
> 1. we decouple the information decomposition and alignment, in the sense that we first preserve task-relevant information as much as possible, then align them between domains.
> 2. In addition, motivated by Grad-CAM, we decompose information into classification-related and classification-irrelevant ones based on the gradients. This way of generating the weights for information decomposition is simple and non-parametric, eschewing complex loss design and additional computational cost.
>
> References
> > [ref 1] Cao, Jinming, Oren Katzir, Peng Jiang, Dani Lischinski, Danny Cohen-Or, Changhe Tu, and Yangyan Li. "Dida: Disentangled synthesis for domain adaptation." arXiv preprint arXiv:1805.08019 (2018).
> >
> > [ref 2] Cai, Ruichu, Zijian Li, Pengfei Wei, Jie Qiao, Kun Zhang, and Zhifeng Hao. "Learning disentangled semantic representation for domain adaptation." In IJCAI, 2019.
> >
> > [ref 3] Peng, Xingchao, Zijun Huang, Ximeng Sun, and Kate Saenko. "Domain agnostic learning with disentangled representations." In ICML 2019.
>
> **Q3: The authors don’t discussion potential negative societal impacts of their work.**
>
> **A3:** Thank you for the nice suggestion. We will discuss the potential negative societal impacts of ToAlign in the Broader Impact section in the revision.
>
> The major negative societal impact of ToAlign arises from the setting of UDA, where only the source domain is annotated, and negative bias of the source domain may be introduced. Thus, we should be especially careful in applying such kind of methodologies to mission-critical tasks, such as medical applications, security, etc.

---

> > ### Comment · Reviewer_9jPd · 2021-08-31
> > **Response (Task vs Class Clarification)**
> >
> > Thank you for the clarification, based upon your response I better understand how your work is positioned in regards to previous disentanglement works and find the omission of the works I mentioned less worrisome.
> >
> > A cause of confusion for me was the use of task for what conceptually maps to a class, in my understanding. In my understanding N-way classification is a single task with N classes, it seems you're presenting this a N tasks to be aligned.
> >
> > I'm updating my scoring from a 6 to 7 in light of your rebuttals.

---

> > > ### Author Response · Authors · 2021-09-01
> > > **Response**
> > >
> > > Thank your for the positive comment on our previous response.
> > >
> > > 1. Yes. N-way classification is a single task with N classes.
> > > 2. In our proposed *ToAlign*, the N-class (N-way) alignment is **jointly learned with one model**. Therefore, we do not
> > > treat the learning of alignment as N independent tasks.

---

> > > > ### Author Response · Authors · 2021-09-02
> > > > **Response to Reviewer 9jPd**
> > > >
> > > >
> > > > Thank you for your positive feedback!
> > > >
> > > > We hope our response could help address your concern. If you have any further comment, we are really glad to help address it here, considering that the deadline of rolling discussion is coming.

---

### Official Review · Reviewer_6xwo · 2021-07-21

**Rating:** 7
**Confidence:** 4

**Summary:**

This paper address the problem of Unsupervised Domain Adaptation. In particular, the authors investigate if aligning only task-oriented features could help to improve the results. They propose an approach that is mainly based on Grad-CAM. They use Grad-CAM to decompose a holistic feature into a task-discriminative feature and a task-irrelevant feature by softly selecting weights for the features. Later these features are integrated with an adversarial approach for domain adaptation. The authors demonstrate improved performance on several datasets. Additionally, in the supplementary material, the authors demonstrate feature response maps for the baseline and proposed approach on the source and target datasets. The method seems to indeed pick task-relevant features even on the target dataset, which was not necessarily the case for the baseline.

**Limitations And Societal Impact:**

The authors clearly discuss limitations in the main and in the supplementary text.

**Main Review:**

I find the proposed approach very interesting and original. The proposed approach is rather a heuristic in nature, but it is well-aligned with the line of works focusing on only aligning features that serve classification problems. To me, it looks like a good attempt to distinguish between causal and non-causal features for classification and align only causal ones. There are some works showing that having causal features would serve generalization (e.g. https://arxiv.org/abs/1907.02893), therefore I find this method for domain adaptation interesting. Even this link be more on an intuitive level in this work and demonstrated theoretically, I find experimental results quite convincing.

**Time Spent Reviewing:**

5

---

> ### Author Response · Authors · 2021-08-10
> **Response to Reviewer 6xwo**
>
> We thank you for the positive comments on the novelty/originality of our work as well as the insightful comments on the understanding of this work.
>
> Actually, the selected task-discriminative information in ToAlign is the causal information for classification which should be aligned with the target domain so as to be free of the influence of the domain gap as much as possible. In this work, we pinpoint it is necessary to find the causal information in source efficiently and then align it with target for better Domain Adaptation. Specifically, we obtain the causal information in a simple and effective way (i.e., using Grad-CAM), then incorporate this information into existing domain alignment procedures.
> IRM (https://arxiv.org/abs/1907.02893) aims to learn the causal (domain-invariant) information with the proposed loss regularization towards better generalization capability, where there is no further operation on this information, e.g., domain alignment.

---

### Official Review · Reviewer_uRc3 · 2021-07-22

**Rating:** 8
**Confidence:** 3

**Summary:**

The paper proposes use of Grad-CAM to identify positively aligned features for domain adversarial training to improve the downstream classification task in the target domain.

**Limitations And Societal Impact:**

The broader impact section is left to Appendix instead of main paper

**Main Review:**

I liked the paper overall. The idea is intuitive, it is well explained and compared against related work in literature. The evaluation is thorough in terms of comparisons with baselines, considering multiple scenarios and performing ablations. The visualization of the features are very useful to understand the concepts presented.

A few suggestions to improve the paper:
- Error bars have been shown in the appendix. It would be nice to have a pointer in the main paper noting that the proposed method is statistically better after considering variance across three runs with details in Appendix.
- The Grad-CAM based methods proposed only applies to vision based models. The way the paper is written, it looks like the proposed solution is more broadly applicable. It will be good to call out this limitation and perhaps propose alternative approaches for other domains.



**Time Spent Reviewing:**

4

---

> ### Author Response · Authors · 2021-08-10
> **Response to Reviewer uRc3**
>
> We thank you for the positive comments on the novelty of the idea and nice suggestions on the writing. Please find our responses below:
>
> **Q1:  Error bars have been shown in the appendix. It would be nice to have a pointer in the main paper noting that the proposed method is statistically better after considering variance across three runs with details in Appendix.**
>
> **A1:** Thank you for this good suggestion. We will add a pointer as suggested to the error bars in Section 4.3 in our revision.
>
> **Q2: The Grad-CAM based methods proposed only applies to vision based models. The way the paper is written, it looks like the proposed solution is more broadly applicable. It will be good to call out this limitation and perhaps propose alternative approaches for other domains.**
>
> **A2:** We appreciate your valuable suggestion. We would like to clarify that ToAlign is not limited to vision-based methods in the revision and point out the potential of extending this idea to other domains in the future work.
> As shown in [45], Grad-CAM is not limited to only vision tasks but also applicable to NLP methods. Therefore, our proposed ToAlign is expected to be also applicable to domain alignment based NLP tasks, e.g., unsupervised language adaptation [ref 1], cross-lingual sentiment classification [ref 2], domain adaptation for machine reading comprehension [ref 3], etc.
>
> References:
>
> > [ref 1] Rocha, Gil, and Henrique Lopes Cardoso. "A comparative analysis of unsupervised language adaptation methods." In Proceedings of the 2nd Workshop on Deep Learning Approaches for Low-Resource NLP, 2019.
> >
> > [ref 2] Chen, Xilun, Yu Sun, Ben Athiwaratkun, Claire Cardie, and Kilian Weinberger. "Adversarial deep averaging networks for cross-lingual sentiment classification." Transactions of the Association for Computational Linguistics, 2018.
> >
> > [ref 3] Wang, Huazheng, Gan, Zhe, Liu, Xiaodong, Liu, Jingjing, Gao, Jianfeng and Wang, Hongning. "Adversarial Domain Adaptation for Machine Reading Comprehension." In EMNLP, 2019.
>
> **Q3: The broader impact section is left to Appendix instead of main paper**
>
> **A3:** We will add broader impact in the main paper in the revision.

---

### Author Response · Authors · 2021-08-10
**General answer to Reviewers**

We sincerely thank all reviewers for the valuable suggestions and positive comments on the novelty, effectiveness, and comprehensiveness of experiments.

---

### Decision · Program_Chairs · 2021-09-27

**Decision:**

Accept (Poster)

**Comment:**

This paper uses Grad-CAM to reweigh the latent features in order to align the source and target domains only with respect to task-oriented features. The method is simple and effective. The idea is clearly motivated in the context of relevant literature and the experimental results clearly demonstrate the effectiveness of the method. All reviewers, including myself, find the paper an interesting and solid contribution that is worth publication.